



# TRAPPIST-1 Habitable Atmosphere Intercomparison (THAI). Motivations and protocol

Thomas Fauchez[1,2,3], Martin Turbet[4,5], Eric T. Wolf[6,7], Ian Boutle[8], Michael J. Way[9,3], Anthony D. Del Genio[9], Nathan J. Mayne[10], Konstantinos Tsigaridis[9,12], Ravi K. Kopparapu[2,3], Jun Yang[11], Francois Forget[4], Avi Mandell[2,3], and Shawn D. Domagal Goldman[2,3]

[1]Goddard Earth Sciences Technology and Research (GESTAR), Universities Space Research Association (USRA), Columbia, Maryland, USA
[2]NASA Goddard Space Flight Center, Greenbelt, Maryland, USA
[3]GSFC Sellers Exoplanet Environments Collaboration
[4]Laboratoire de Méteorologie Dynamique, IPSL, Sorbonne Universités, UPMC Univ Paris 06, CNRS, 4 Place Jussieu, 75005 Paris, France
[5]Observatoire Astronomique de l'Université de Genève, Université de Genève, Chemin des Maillettes 51, 1290 Versoix, Switzerland.
[6]Laboratory for Atmospheric and Space Physics, Department of Atmospheric and Oceanic Sciences, University of Colorado Boulder, Boulder, CO, USA
[7]NASA Astrobiology Institute's Virtual Planetary Laboratory, Seattle, WA, USA
[8]Met Office, Exeter, UK
[9]NASA Goddard Institute for Space Studies, New York, NY 10025, USA
[10]University of Exeter, Exeter, UK
[11]Dept. of Atmospheric and Oceanic Sciences, School of Physics, Peking University, Beijing, 100871
[12]Center for Climate Systems Research, Columbia University, New York, NY, USA.

**Correspondence:** Thomas J. Fauchez (thomas.j.fauchez@nasa.gov)

**Abstract.** Upcoming telescopes such as the James Webb Space Telescope (JWST), the European Extremely Large Telescope (E-ELT), the Thirty Meter Telescope (TMT) or the Giant Magellan Telescope (GMT) may soon be able to characterize, through transmission spectroscopy, the atmospheres of rocky exoplanets orbiting nearby M dwarfs. One of the most promising candidates is the late M dwarf system TRAPPIST-1 which has seven known transiting planets for which Transit Timing Variation (TTV) measurements suggest that they are terrestrial in nature, with a possible enrichment in volatiles. Among these seven planets, TRAPPIST-1e seems to be the most promising candidate to have habitable surface conditions, receiving ∼ 66% of the Earth's incident radiation, and thus needing only modest greenhouse gas inventories to raise surface temperatures to allow surface liquid water to exist. TRAPPIST-1e is therefore one of the prime targets for JWST atmospheric characterization. In this context, the modeling of its potential atmosphere is an essential step prior to observation. Global Climate Models (GCMs) offer the most detailed way to simulate planetary atmospheres. However, intrinsic differences exist between GCMs which can lead to different climate prediction and thus observability of gas and/or cloud features in transmission and thermal emission spectra. Such differences should preferably be known prior to observations. In this paper we present a protocol to inter-compare planetary GCMs. Four testing cases are considered for TRAPPIST-1e but the methodology is applicable to other rocky exoplanets in the Habitable Zone. The four test cases included two land planets composed of pure $N_2$ and pure $CO_2$, respectively, and two



aqua planets with a modern Earth and a $CO_2$ rich composition. Currently there are 4 participating models (LMDG, ROCKE3D, ExoCAM, UM), however this protocol is intended to let other teams participate as well.

# 1 Introduction

M dwarfs are the most common type of stars in our galaxy and rocky exoplanets orbiting M dwarf stars will likely be the
first to be characterized with upcoming astronomical facilities such as the James Webb Space Telescope (JWST). Ultra-cool dwarfs (T < 2700 K) are a sub-stellar class of late M-dwarfs and represent nearly $20\%$ of astronomical objects in the stellar neighborhood of the Sun. Their smaller size compared to other stellar types allows easier detection of rocky exoplanets in close orbits, and this potential was recently realized by the discovery of the TRAPPIST-1 system (Gillon et al., 2016, 2017). Located about 12 pc away TRAPPIST-1 has seven known planets, and is one of the most promising rocky-planet systems for
follow-up observations due to the depths of the transit signals (Gillon et al., 2017; Luger et al., 2017). Transit Timing Variation (TTVs) measurements of the TRAPPIST-1 planets suggest a terrestrial composition likely enriched in volatiles, and possibly water (Grimm et al., 2018). Also, it has been found that three planets (TRAPPIST-1 e, f and g) are in the habitable zone (HZ, Kopparapu et al., 2013) where surface temperatures would allow surface water to exist (Gillon et al., 2017; Wolf, 2017, 2018; Turbet et al., 2018).

TRAPPIST-1 is an active M dwarf star (O'Malley-James and Kaltenegger, 2017; Wheatley et al., 2017; Vida and Roettenbacher, 2018) which offers an environment very hostile to the survival of planetary atmospheres. However, Bolmont et al. (2017) and Bourrier et al. (2017) argued that depending on their initial water contents, the TRAPPIST-1 planets could have retained some water presently. Assuming that this water has remained in sufficient quantity, TRAPPIST-1e may be able to maintain habitable conditions (locally or globally around the planet) through a very large set of atmospheric configurations
(Wolf, 2017; Turbet et al., 2018; Grootel et al., 2018, and references therein). The first attempt to characterize those planets through transmission spectroscopy has been conducted by de Wit et al. (2016, 2018) using the Hubble Space Telescope (HST) for the six innermost planets. Their analysis suggests that the TRAPPIST-1 planets do not have a cloud/haze free $H_2$ dominated atmosphere and that a large set of high mean molecular weight atmospheres are possible, such as thick $N_2$, $O_2$, $H_2O$, $CO_2$, or $CH_4$ dominated atmospheres. Using laboratory measurements and models Moran et al. (2018) have also shown that
$H_2$ dominated atmospheres with cloud/haze can also be ruled out. Note that the uncertainties of these HST observations were very large, on the order of hundreds of parts per million (ppm) and further investigations with future facilities such as JWST (Barstow and Irwin, 2016; Morley et al., 2017) will be needed to determine the nature of atmospheres heavier than hydrogen.

Upstream of future JWST characterization of TRAPPIST-1e, it is important to derive constraints on its possible atmosphere to serve as a guideline for the observations. For this purpose, 3-D Global Climate Models (GCMs) are the most advanced tools
(Wolf et al., 2019). However, GCMs are very complex models and their outputs can vary from one model to another for a variety of reasons. GCM intercomparisons have been widely used by the Earth science community. For instance the Coupled Model Intercomparison Project (CMIP) initiated in 1995 and currently in its version 6 (Eyring et al., 2016), focuses on the differences in GCM responses to forcings for anthropogenic climate change. While exoplanets receive considerable attention from



climate modelers, and atmospheric data from Earth-like worlds may be imminent, to our knowledge only one intercomparison of planetary GCMs has been published (Yang et al., 2019). They found significant differences in global surface temperature between the models for planets around M-dwarf stars due to differences in atmospheric dynamics, clouds and radiative transfer. However, Yang et al. (2019) concerns planets near the inner edge of the HZ and focuses on highly idealized planetary

configurations. Note that another model intercomparison have been run for the exoplanet community: the Palaeoclimate and Terrestrial Exoplanet Radiative Transfer Model Intercomparison Project (PALAEOTRIP). The protocol of this experiment is described in Goldblatt et al. (2017) and aims to compare a large variety of radiation codes used for paleoclimate or exoplanets sciences, to identify the limit conditions for which each model can produce accurate results. Information and timeline about PALEOTRIP can be found at http://www.palaeotrip.org/.

The motivation behind the TRAPPIST Habitable Atmosphere Intercomparison (THAI), is to highlight differences among GCM simulations of a confirmed exoplanet, TRAPPIST-1e, that is potentially characterizable in the near term (with JWST or ground-based facilities), and to evaluate how these differences may impact our interpretations of retrievals of its atmospheric properties from delivered observables. Our objective is also to provide a clear protocol intended for other GCMs to join the intercomparison, which is therefore not only limited to the GCMs presented in this paper. Results of the intercomparison will be presented

in a following paper. In this paper, the motivations, including a presentation of TRAPPIST-1e and of the GCMs, are presented in section 2. In section 3 we present the THAI protocol describing all the parameters to be set up in the GCM. In Section 4, we list the model parameters to be provided in order for a given model to be comparable to other GCM simulations. A summary is given in section 5.

## 2   TRAPPIST-1e climate simulation and motivations

### 2.1   Motivations for a planetary GCM intercomparison

Global Climate Models (GCMs) are 3-dimensional numerical models designed to represent physical processes at play in planetary atmospheres and surfaces. They are the most sophisticated way to model the atmospheres and oceans of real planets. GCMs can be seen as a complex network of 1-D time-marching climate models connected together through a dynamical core (see description below). Each 1-D column contains physical parameterizations for radiative transfer, convection, boundary

layer processes, cloud macroscale and microscale physics, aerosols, precipitation, surface snow and sea ice accumulation, and other processes, at varying levels of complexity.

The motivation behind this experimental protocol is to evaluate how some of the differences between the models can impact the assessment of the planet's habitability and its observables through transmission spectroscopy and thermal phase curves with upcoming observatories such as JWST. The intercomparison protocol was designed to evaluate three possible sources of

differences between the models listed below:

1. The dynamical core:

    The dynamical core is a numerical solver of the hydrodynamic equations on the (rotating) planetary sphere. It calcu-





lates the winds that transport atmospheric gases, clouds, aerosols, sensible and latent heat, and momentum from one atmospheric column to another.

2. The radiative transfer:

Each model has its own radiative transfer working assumptions and may use different spectroscopic databases and even different versions of the same spectroscopic database (e.g., HITRAN), collision-induced absorption (CIA), line-by-line versus correlated-k distribution (Lacis and Oinas, 1991), line cutoff, spectral resolution, etc.

3. The moist physics:

The treatment of water in all of its thermodynamic phases is critical for the simulation of habitable planets. In particular cloud and convection process are a significant source of differences between climate models, and these differences are often exacerbated when modeling exoplanets around M-dwarf stars (Yang et al., 2014, 2019).

Note that a particular emphasis will be given on the differences of cloud properties between the models because they may have a large impact on the strength of the spectral signatures simulated by current radiative transfer tools (Fauchez et al., 2019). Yet a sufficient understanding of 3D cloud fields is needed to provide realistic observational constraints to observers. It is therefore crucial to address these potential differences between the GCMs.

Four GCMs (in their planetary version) are initially onboard THAI:

1. the Laboratoire de Météorologie Dynamique - Generic model (LMDG, Wordsworth et al., 2011, a review paper on the model is currently under preparation),

2. the Resolving Orbital and Climate Keys of Earth and Extraterrestrial Environments with Dynamics (ROCKE3D, Planet 1.0 version derived from the NASA GISS Model E, Way et al., 2017),

3. the Exoplanet Community Atmospheric Model (ExoCAM [1], derived from the CAM4 NCAR model, Neale, 2010),

4. the Met Office Unified Model (UM, Mayne et al., 2014; Boutle et al., 2017).

By publishing our protocols in advance of the intercomparison work, we hope that other teams will also use this protocol to compare their own GCM with the four GCMs of this study.

## 2.2 The TRAPPIST-1e benchmark

TRAPPIST-1e is up to now one of the best habitable planet candidates for atmospheric characterization through transmission spectroscopy with JWST. Therefore, it is also an obvious candidate for an experimental protocol for GCM intercomparison. In Table 1 we summarize the TRAPPIST-1e parameters used in the THAI project based on Grimm et al. (2018).

---

[1]Available on Github, https://github.com/storyofthewolf/ExoCAM

Available from NCAR, http://www.cesm.ucar.edu/models/cesm1.2/

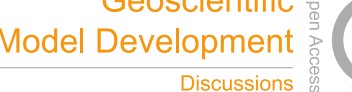

**Table 1.** TRAPPIST-1 stellar spectrum and TRAPPIST-1e planetary parameters from Grimm et al. (2018)

| Star & spectrum | 2600 K BT Settl with Fe/H = 0 |
|---|---|
| Planet | TRAPPIST-1e |
| Insolation | $900\ W.m^{-2}$ |
| Rotation period | 6.1 days |
| Orbital period | 6.1 days |
| Mass ($M_{\oplus}$) | 0.772 |
| Radius ($R_{\oplus}$) | 0.910 |
| Density ($\rho_{\oplus}$) | 1.024 |
| Gravity ($g_{\oplus}$) | 0.930 |

## 3 The THAI Protocol

### 3.1 Atmospheric configurations

For THAI, we have chosen a set of four planetary configurations with increasing complexity. We have chosen to start with benchmark cases of dry-land planets with $N_2$- and $CO_2$-dominated atmospheres respectively, which will allow us to assess
5   atmospheric dynamical + boundary layer, and $CO_2$ radiative transfer differences. Next we conduct aquaplanet simulations of $N_2$ and $CO_2$ dominated atmospheres respectively, providing characteristic cold and warm habitable states for TRAPPIST-1e. By gradually increasing the complexity of our simulations, we hope to be able to parse out meaningful differences between atmospheric dynamical + boundary layer, radiative transfer, and moist physical processes. The motivation for each of these cases is described below:

10   – Benchmark case 1 (Ben1): In this case, constituted of 1 bar of $N_2$ only, the purpose is to test the differences of the planetary boundary layer (PBL) schemes, the dynamical core and the associated heat redistribution between the different models.

  – Benchmark case 2 (Ben2): In this case, constituted of 1 bar of $CO_2$, we test the PBL schemes and dynamical core differences as well as the $CO_2$ radiative transfer.

15   – Habitable case 1 (Hab1): In this case, constituted of a modern Earth-like atmosphere of 1 bar of $N_2$ and 400 ppm of $CO_2$, the dynamical core, the clouds and atmospheric processes are tested together. It is also the most widespread benchmark for habitable planets in the literature (Barstow and Irwin, 2016; Morley et al., 2017; Lincowski et al., 2018).

  – Habitable case 2 (Hab2): In this case, constituted of 1 bar of $CO_2$, the dynamical core, the $CO_2$ radiative transfer assumption and the clouds and atmospheric processes are tested. This case is likely representative of the early Earth





(during the Hadean epoch), early Venus, and early Mars, at a time when Martian valley networks and lakes were formed (Haberle et al., 2017; Kite, 2019).

In each case, it is crucial to start each simulation with the same initial conditions. The simplest approach is then to start with an isothermal atmosphere. For THAI, we fixed the initial surface and atmosphere temperature at 300 K. The atmospheric configurations for the two benchmark (dry land) cases and two habitable cases are listed in Table 2, first horizontal block. Note that for Ben2 initial results indicate that some models feature cold trap temperatures on the night-side slightly below the $CO_2$ condensation point at 1 bar (194 K). However, because all the models do not include $CO_2$ condensation it should be disabled in the models that allow it. Ben2 is thus to be viewed as a idealization for the sake of study. Initial results indicate that Hab1 is representative of a cool, largely ice covered world but with liquid water in the substellar region. Hab2 is significantly warmer than Hab1, owing to a strong $CO_2$ greenhouse effect and the water vapor greenhouse feedback, and is representative of a temperate habitable world. The amount and variability of clouds and the strength of the atmospheric processes should be enhanced providing a more challenging comparison than in Hab1.

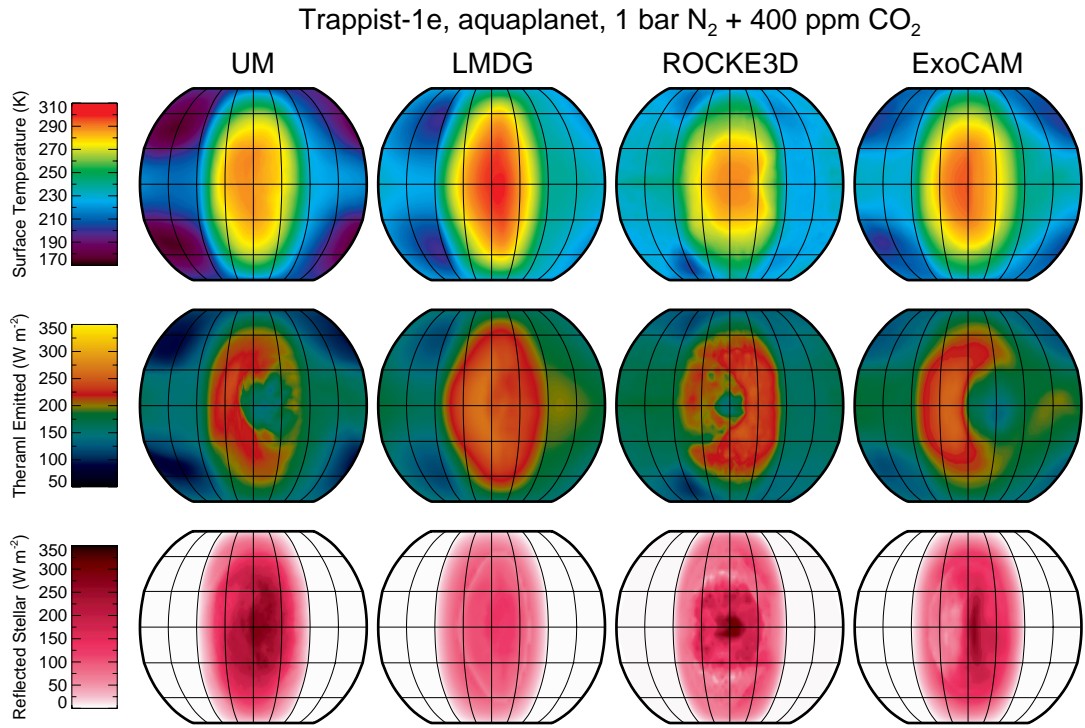

**Figure 1.** Surface contours for surface temperature, thermal emitted radiation (TOA) and reflected stellar radiation (TOA) for "Hab1" simulated by the four GCMs: the UK Met Office United Model (UM), the Laboratoire Météorologie Dynamique Generic model (LMDG), the Resolving Orbital and Climate Keys of Earth and Extraterrestrial Environments with Dynamics (ROCKE3D), and the National Center for Atmospheric Research Community Atmosphere Model version 4 modified for exoplanets (ExoCAM).





In Fig. 1 we show results from preliminary simulations on case "Hab1" conducted with four different GCMs; UM, LMDG, ROCKE3D and ExoCAM. We show surface contours for surface temperature, thermal emitted radiation (TOA) and reflected stellar radiation (TOA). We can see significant differences in the maximum, minimum and mean values of these parameters between the models. For such a complex atmosphere it is difficult to disentangle the effects leading to these differences. However, it seems clear that the patterns of thermal emitted and reflected stellar TOA fluxes are strongly influenced by the cloud patterns produced by each respective model. Here we have shown preliminary outputs to demonstrate the feasibility of the described experiments. In depth analysis of these simulations will be discussed in a following manuscript in preparation.

## 3.2 Surface

The surfaces considered in THAI (Table 2, second horizontal block) are simple. The land planets (Ben1 & Ben2) are covered by sand with a subsurface depth of at least 3 m with a constant albedo of 0.3. The ocean planets (Hab1 & Hab2) are fully covered by a 100 m deep slab (no horizontal heat transport) ocean. The ocean albedo is fixed at 0.06 and the ice and snow albedos are fixed at a constant value of 0.25. Note that the sea ice/snow albedo parameterization is a common source of discrepancy between the models. Some models, like ROCKE3D, account for the spectral dependence of the sea ice albedo over multiple bands and variations due to snowfall, aging, depth and melt ponds while other models, such as LMDG, compute the wavelength-dependent albedo of water ice / snow from a simplified albedo spectral law, calibrated to get an ice / snow bolometric albedo of $\sim 0.25$ around an ultra-cool star like TRAPPIST-1 (Joshi and Haberle, 2012; von Paris et al., 2013; Shields et al., 2013). Differences in sea ice albedo have been found to have a large impact on planetary climate and habitability (Turbet et al., 2018). However, for the sake of this intercomparison, this discrepancy can be easily avoided by fixing the sea ice and snow albedo at a constant bolometric value of 0.25.

## 3.3 Model spatial resolutions and time steps.

The model spatial resolution is an important parameter because every process taking place at a sub-grid level would be parameterized and those parameterizations often diverge between the models. Similarly the model time steps control the numerical stability and accuracy. However, the choices for those are fundamental to how each model operates under a given parameterization and arbitrary fixing these parameters may prevent some model to correctly and fairly perform the intercomparison. In addition, models should be compared using the specifications that they commonly use for exoplanet studies. Therefore, for the sake of the THAI, we do not impose the model spatial resolution nor time steps. Note that we however recommend (but this is not a requirement) the radiative time step (a parameter much more flexible than the others among the models) to be set up at 1800 s. This value should provide a good coupling of the radiation with temporal changes to the atmosphere without slowing down to much the model.

We also ask the contributing scientists to disable the gravity waves in their model. Indeed, all the models do not have implemented a gravity wave parameterization and some have prescribed or predicted gravity wave formation, tuned for Earth topography and meteorology. Therefore, to avoid differences in the mesosphere dynamics, we recommend to not include grav-





ity waves in this intercomparison.

Note that under the requirements of the protocol, the atmospheric simulation of TRAPPIST-1e may actually not represent what each individual model can simulate with all their parameterizations fully activated. This is especially true for the sea ice and snow albedo parameterization. Therefore, complementary to the Hab1 case, we propose the Hab1* which should be simulated with the commonly used model parameterizations fully activated. Therefore, only the requirements on the atmospheric composition (1 bar of $N_2$ and 400 ppm of $CO_2$) and the planet and star properties of Table 1 are constrained for Hab1*.

**Table 2.** THAI experimental protocol.

| Case | Ben1 | Ben2 | Hab1 | Hab2 |
|---|---|---|---|---|
| **Atmospheres** | | | | |
| Composition | 1 bar $N_2$ | 1 bar $CO_2$ | 1 bar $N_2$ + 400 ppm $CO_2$ | 1 bar $CO_2$ |
| Molecular air mass | 28 | 44 | 28 | 44 |
| Initial state | Isothermal 300 K | | Isothermal 300 K | |
| **Surfaces** | | | | |
| Type | Land only | | Ocean planet | |
| Composition | Sand | | Slab ocean | |
| Albedo | 0.3 | | Liquid water: 0.06 Ice/snow: 0.25 | |
| Heat capacity ( $J/m^3/K$ ) | $2 \cdot 10^6$ | | $4 \cdot 10^6$ | |
| Thermal inertia ( $J/m^2/K/s^2$ ) | 2000 | | 12000 | |
| Momentum roughness length | 0.01 | | 0.01 | |
| Heat roughness length | 0.001 | | 0.001 | |
| Depth of the subsurface / ocean | > 3 m | | 100 m | |
| Cautions: | disable gravity waves disable $CO_2$ condensation | | | |

## 4 Outputs

To compare the difference between models of a particular (instantaneous) output variable, both the average and standard deviation over the specified frequency and number of orbits for the case will be computed. Four categories of outputs frequently used in climate simulations have been selected: radiation, surface, atmospheric profiles and clouds. The radiation outputs are the outgoing longwave radiation (OLR) and absorbed shortwave radiation (ASR) for clear and cloudy skies, also commonly



known as emitted thermal and absorbed stellar fluxes, respectively, both at the top of the atmosphere (TOA). The surface outputs are the temperature map, the downward total SW flux and net LW flux and the open ocean fraction (for Hab1/Hab1* & Hab2 only). The atmosphere outputs are the temperature and the U, V and W wind speed profiles. Finally, the cloud outputs For Hab1/Hab1* & Hab2 are the water vapour and cloud condensed water and ice integrated columns, and the cloud profiles

5 of the cloud fraction and the mass mixing ratio for the liquid, ice and both combined. Also in these two cases, the spatial and temporal variability is much weaker than in Hab1/Hab1* & Hab2. Therefore, to mitigate the amount of data we choose to only output data for ten consecutive orbits (with a 6 hour output frequency). Concerning Hab1/Hab1* & Hab2, we can see in Figure 2 that weather patterns modulate the surface temperature and cloud water column of Hab1 on a period nearly equal to 10 orbits. Also Hab2 (hotter than Hab1) has a more efficient heat transport and is therefore more homogeneous in temperature but the

10 cloud variability is very important. Therefore, more orbits (100) are needed in order to smooth out this variability. A summary of the output parameters is given in Table 3.

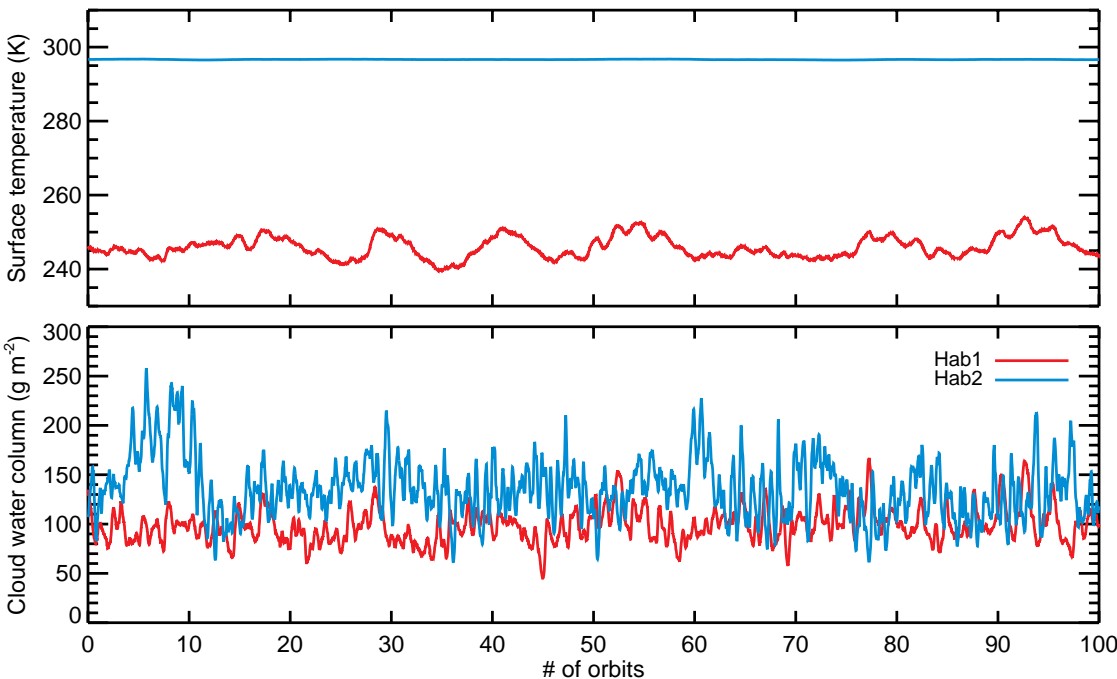

**Figure 2.** Globally averaged surface temperature (top panel) and cloud water column (bottom panel) as a function of the number of orbits for Hab1 & Hab2 simulated with ExoCAM (Wolf and Toon, 2015). Surface temperature for Hab1 and cloud water column for the 2 cases vary by a couple of tens of percents on a timescale of 10 orbits due to weather patterns.

All the simulations should have reached radiative equilibrium at TOA. To facilitate comparison between each GCM, we ask the contributing scientists to provide their output in netCDF format.





**Table 3.** Instantaneous fields to be output by the GCM. For each diagnostic, the mean value and the standard deviation are computed from data output at the specified frequency and number of orbits for the case. OLR and ASR correspond to outgoing longwave radiation (at TOA) and absorbed shortwave radiation (at TOA), respectively, SW and LW correspond to shortwave and longwave, respectively, CF corresponds to cloud fraction and MMR at mass mixing ratio.

| Case | Ben1 | Ben2 | Hab1/Hab1* | Hab2 |
|---|---|---|---|---|
| Number of orbits | 10 | 10 | 100 | 100 |
| Frequency (hours) | 6 | 6 | 6 | 6 |
| **2D maps** | | | | |
| Radiation | | | OLR (clear/cloudy) | |
| | | | ASR (clear/cloudy) | |
| Surface | | | temperature map | |
| | | | downward total SW flux | |
| | | | Net LW flux | |
| | ∅ | ∅ | open ocean fraction | |
| Clouds | ∅ | ∅ | total/liquid/ice/vapor column | |
| **Vertical profiles** | | | | |
| Atmospheric | | | temperature | |
| profiles | | | U, V , W wind speed | |
| | | | heating rates (SW/LW) | |
| | ∅ | ∅ | specific + relative humidity | |
| Cloud profiles | ∅ | ∅ | CF (total/liquid/ice) [%] | |
| | ∅ | ∅ | MMR (total/liquid/ice) [kg/kg] | |

The main objective of THAI is to highlight how differences in atmospheric profiles produced by each GCM are going to impact the predictions of atmosphere detectability and observational constraints for habitable planet targets such as TRAPPIST-1e (Morley et al., 2017; Fauchez et al., 2019). Therefore, in addition to the parameters of Table 3, we will emphasize the differences between the models in term of the planet's climate and habitability with a particular attention on the cloud coverage.

5   Also, the objective will be to identify and quantify the differences on the simulated JWST observations, through simulated transmission spectra (in NIRSpec prism and MIRI LRS ranges) and thermal phase curves (in MIRI LRS range) due to the differences of atmospheric profiles (temperature, pressure and gas mixing ratios) output by each GCM. The planetary spectrum generator (PSG, Villanueva et al. (2018)) will be used to simulate transmission and emission spectra. The comparison of the spectra for Hab1 & Hab2 cases will therefore highlight the sensitivity of model characteristics to predict JWST transmission

10   spectra of habitable planets.



## 5 Summary

THAI is an intercomparison project of planetary GCMs focused on the exciting new habitable planet candidate, TRAPPIST-1e. Because rocky exoplanets in the Habitable Zone of nearby M dwarfs have the highest chance to be the first Earth-size exoplanets to be characterized with future observatories, TRAPPIST-1e is currently the best benchmark we could think of to compare the capability of planetary GCMs. In this first paper we have presented the planet and GCM parameters to be used in this experiment which already has four GCMs onboard (LMDG, ROCKE-3D, ExoCAM and UM), but we hope more GCMs will join the project. The results of the comparison of these four models will be given in a second paper.

*Code availability.* ExoCAM (Wolf and Toon, 2015) is available on Github, https://github.com/storyofthewolf/ExoCAM. The Met Office Unified Model is available for use under licence, see http://www.metoffice.gov.uk/research/modelling-systems/unified-model.ROCKE-3D is public domain software and available for download for free from https://simplex.giss.nasa.gov/gcm/ROCKE-3D/. Annual tutorials for new users take place annually, whose recordings are freely available on line at https://www.youtube.com/user/NASAGISStv/playlists?view=50&sort=dd&shelf_id=15. LMD-G is available upon request.

*Data availability.* No data have been used in this study

*Competing interests.* No competing interests are present.

## Author contribution

T.J.F. lead the THAI project and has written the manuscript. E.T.W ran the simulation for Fig 1. and plotted the figures. Every author contributed to the development of the THAI protocol, to the discussions and to the editing of the manuscript.

*Acknowledgements.* Goddard affiliates are thankful for support from GSFC Sellers Exoplanet Environments Collaboration (SEEC), which is funded by the NASA Planetary Science Divisions Internal Scientist Funding Model. M.T. acknowledges the use of the computing resources on OCCIGEN (CINES, French National HPC).

This project has received funding from the European Union's Horizon 2020 research and innovation program under the Marie Sklodowska-Curie Grant Agreement No. 832738/ESCAPE. M.W. and A.D. acknowledge funding from the NASA Astrobiology Program through participation in the Nexus for Exoplanet System Science (NExSS).

The THAI GCM intercomparison team is grateful to the Anong's THAI Cuisine restaurant in Laramie for hosting its first meeting on November 15, 2017.





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
