# Peer review of "TRAPPIST-1 Habitable Atmosphere Intercomparison (THAI). Motivations and protocol"

_Geoscientific Model Development, 2019_

## Referee Comment (RC1) · Daniel D.B. Koll (Referee) · 25 Aug 2019

This is an interesting proposal for a exoplanet GCM (global climate model) intercomparison. I am a fan of the general idea and the project's open nature – given the booming interest in the field, model intercomparisons such as this one will be playing an important ongoing role. I also think the experimental setup is generally appropriate, but clarifications on the overall design and on some of the modeling choices would be helpful. See below.

*Major comments: - What plans do the authors have for an online presence to share necessary input files as well as model outputs? In particular, it is not clear to me how participants who are interested in submitting a model would obtain the necessary host

star spectrum. I would also strongly encourage the authors to make their main output netcdfs publicly available (=not on a personal/professional website) so that model developers or graduate students will still be able to access the results five years from now. Github sounds like an easy option. Similarly some of the co-authors, e.g., the ROCKE3D team, have been doing great work in making their files available to the rest of the community, which might also be feasible here.

- To understand the likely-important impact of cloud parameterizations, what about a version of Hab1/Hab2 that still includes water vapor/latent heating effects but disables the radiative effects of clouds (see Yang et al 2019)? This setup should be easy to implement in most GCMs.

- I know that this is not easily done with many GCMs, but adding a 1D single column case to the intercomparison would be very useful for isolating differences due to clearsky radiative transfer. These differences can be far from negligible (see Yang et al, 2016), and at least some of the models in this study should be able to run in 1D. Even if a 1D intercomparison isn't feasible here, calling for such an option would at least be a useful sign to model developers.

- For the pure N2 case, do the authors still want models to include N2-N2 collision-induced absorption or is this supposed to be an atmosphere that is completely transparent? For an intercomparison, the latter case would presumably be interesting. However, I'd think that a zero-opacity atmosphere might easily lead to numerical issues (first, some radiation codes just crash if run with zero optical thickness; second, a zero- or low-opacity atmosphere might become extremely warm, because it is still heated by sensible heat fluxes on the dayside but can't easily shed the heat via radiative cooling, leading to potential further numerical issues). If such issues arose during the study, it would be worth discussing how participating models have dealt with them.

*Minor comments: - Figure 1, bottom row: add global-mean albedos somewhere on this figure?

- Page 7: the intercomparison fixes albedos, but what about sea ice dynamics?

- Page 7: do the Hab1 and Hab2 cases with a zero-ocean-heat-transport slab ocean reach steady state on the nightside? In particular, does the sea ice thickness asymptote to a finite value?

- Page 9 "should have reached radiative equilibrium" To what precision, in W/m2? Also, the global-mean top-of-atmosphere radiative equilibrium will be dominated by the warm dayside. The nightside could take much longer to reach equilibrium (smaller flux = longer equilibration timescale). Have the authors looked at the nightside surface budget, to see if it reaches equilibrium?

- Page 11 "LMD-G is available upon request." From whom?

*Technical comments: - Abstract, l.3: "... may soon be able to characterize, through transmission spectroscopy, the atmospheres of rocky exoplanets..." Why emphasize transits over other techniques that the manuscript mentions later on (e.g., emission spectra or phase curves)? The results of this work will be interesting more broadly.

- Abstract, l.14 "The four test cases included two land planets composed of pure N2 and pure CO2, respectively..." ... pure N2 and pure CO2 *atmospheres*, ...

- P2, l.6-7: "... and represent nearly 20% of astronomical objects in the stellar neighborhood of the Sun. " Interesting! Citation?

- P6, l.7: "because all the models do not include CO2 condensation" - because not all the models include X, or because all the models do not include X?

- P7, l.29: " to much the model" - typo, too.

- P7, l. 30: "disable the gravity waves" - the gravity wave parameterization in the stratosphere? The dynamical cores should still be resolving some internal gravity waves.

- Page 8, Table 2: molecular air mass is referring to the dry background gases only?

**GMDD**

- Page 8, Table 2: momentum roughness length and heat roughness length are missing units.
* * *

---

## Referee Comment (RC2) · Anonymous Referee #2 · 26 Sep 2019

This brief paper describes a protocol for inter comparing GCMs for TRAPPIST-1e in anticipation of future observations. The goal is to determine the differences in climate states when the models are run under similar configurations. These states can be related to spectra or thermal phase curves anticipated from future observations such as JWST. Four GCMs have signed on so far but only preliminary results are available; more detailed analysis will follow. This is a good idea and should be a useful effort.

The biggest uncertainty is the mass and composition of the atmosphere. The only constraint seems to be that HST observations do not favor an extended $H_2$ atmosphere for TRAPPIST-1e. Thus, heavier atmospheres consisting mainly of $N_2$ and $CO_2$ are considered. The models are configured to sort out the effects of the dynamical core, physical packages, and moist processes. This is achieved by comparing four different

runs for each model with different surface and atmospheric conditions. The approach seems reasonable and should give the authors a good start on what will surely be a challenging but stimulating research project.

The authors might consider a few things.

1. If the goal is to determine the differences in model climate states with similar run configurations, it is not clear how model numerics will be separated from model physics. Different dynamical cores running at different resolutions with different numerical schemes will produce different climates. How does one distinguish these differences from those due to real physical processes? I think BEN1/BEN2 should get at some of this but not all of it. Perhaps one way is to run with simple Newtonian cooling using a common relaxation field and time constant. 2. Once that is clarified then an even more daunting task is to isolate changes due to different physics prescriptions. Is the intent to go to that level of detail or to describe what the differences are without analyzing the reasons? Some brief discussion about this would be helpful. 3. With suppressed $CO_2$ condensation nightside surface temperatures are likely to be much warmer than when condensation is included. Without latent heat release atmospheric temperatures will cool and the surface must warm to maintain energy balance. Feedbacks related to moist processes may be affected and this may complicate the interpretation. 4. The radiative effects of clouds, both $CO_2$ and $H_2O$, can be very different between models. Runs with passive clouds might help isolate those effects. Of course, this adds to the analysis work (as does running with Newtonian cooling), but it is a point worth considering. 5. The authors hope to add more models into the mix which will increase the workload. Recognizing that this is not a proposal, it still begs the question of having adequate support and manpower to do the work. Is there?

---

## Author Comment (AC1) · 14 Oct 2019

**dkoll@mit.edu**

**This is an interesting proposal for a exoplanet GCM (global climate model) intercomparison. I am a fan of the general idea and the project's open nature – given the booming interest in the field, model intercomparisons such as this one will be playing an important ongoing role. I also think the experimental setup is generally appropriate, but clarifications on the overall design and on some of the modeling choices would be helpful. See below.**

We would like to thank the reviewer Daniel D.B Koll for his kind words and his helpful and insightful review of our manuscript. We have deeply considered the points brought up by the referee in the revisions to paper. Our reply is detailed below.

**\*Major comments: - What plans do the authors have for an online presence to share necessary input files as well as model outputs? In particular, it is not clear to me how participants who are interested in submitting a model would obtain the necessary host star spectrum. I would also strongly encourage the authors to make their main out- put netcdfs publicly available (=not on a personal/professional website) so that model developers or graduate students will still be able to access the results five years from now. Github sounds like an easy option. Similarly some of the co-authors, e.g., the ROCKE3D team, have been doing great work in making their files available to the rest of the community, which might also be feasible here.**

We have set up this repository hosted at NASA: https://thai.emac.gsfc.nasa.gov/dataset/thai

The outputs from all the models in the intercomparison will be hosted there. The stellar spectrum used will only be in the dataset. Data will be accessible to download by anyone, and upload will be possible with authorized IP address for scientists whose want to contribute to the intercomparison with their own GCM.

We have added a statement about this in the revised manuscript:

*"THAI model outputs and the TRAPPIST-1 stellar spectrum will be progressively uploaded during the intercompaison and will be available at : https://thai.emac.gsfc.nasa.gov/dataset/thai"*

**- To understand the likely-important impact of cloud parameterizations, what about a version of Hab1/Hab2 that still includes water vapor/latent heating effects but disables the radiative effects of clouds (see Yang et al 2019)? This setup should be easy to implement in most GCMs.**

This is a very interesting suggestion that will help to interpret our differences due to cloud physical processes without having the radiative effects of clouds. However, our objective is not to fully understand all the differences between the models but to understand how the first order differences impact the observables from synthetic spectra. The co-authors have debated considerably to arrive at the five configuration chosen for this intercomparison. At this point we would rather not add more required configurations to this intercomparison. However, time permitting amongst participating parties, we encourage the exploration of different configurations and parameters not explicitly include at present.

We add this statement before the conclusion:

*"Note that while additional simulations with a simple Newton cooling model, a 1-D column model, or with cloud radiative effects disabled would help to better understand the differences due to the dynamical cores and cloud physics, they will also dramatically increase the computationnal time, amount of data and effort. Yet, THAI aims to be easily reproducible and not time consuming in order to reach many GCM user groups. The five simulations propose in THAI should be enough to understand the main differences between the GCMs and their impact on the observable. THAI could also be used as a benchmark for a future GCM intercomparison that will specifically aim to understand the finest differences between the models."*

Also a THAI workshop is currently being planned around fall 2020 to discuss about THAI results and their perspectives.

**- I know that this is not easily done with many GCMs, but adding a 1D single col- umn case to the intercomparison would be very useful for isolating differences due to clearsky radiative transfer. These differences can be far from negligible (see Yang et al, 2016), and at least some of the models in this study should be able to run in 1D. Even if a 1D intercomparison isn't feasible here, calling for such an option would at least be a useful sign to model developers.**

We agree with referee #1 that comparison using a 1D single column model could be useful to isolate difference due to clearsky radiative transfer. However, all GCMs do not have a 1d single column version. Also the cloud-free cases Ben1/Ben2 can also allow such radiative transfer comparison.

**- For the pure N2 case, do the authors still want models to include N2-N2 collision- induced absorption or is this supposed to be an atmosphere that is completely trans- parent? For an intercomparison, the latter case would presumably be interesting. How- ever, I'd think that a zero-opacity atmosphere might easily lead to numerical issues (first, some radiation codes just crash if run with zero optical thickness; second, a zero- or low-opacity atmosphere might become extremely warm, because it is still heated by sensible heat fluxes on the dayside but can't easily shed the heat via radiative cooling, leading to potential further numerical issues). If such issues arose during the study, it would be worth discussing how participating models have dealt with them.**

Yes we have included N2-N2 CIA in our simulations otherwise the atmosphere is indeed transparent and numerical issue arise. Also, we have noticed that when lacking an efficient radiative coolant in the high atmosphere like $CO_2$, $N_2$-$N_2$ mid-IR absorption warms the stratosphere and creates an inversion. The figure below shows a comparison between LMD-G, ROCKE-3D and ExoCAM average temperature profiles. In the left panel mid-IR $N_2$-$N_2$ was

omitted in LMD-G, but included in the two other models. In the right figure $N_2$-$N_2$ has been included in the three models and we can see the temperature inversion occurring between 30-100 mb.

[Figure]

In section 3.1, when presenting Ben2 we have added: *"Note that $N_2$ -$N_2$ CIA should be included to avoid a fully transparent atmosphere and associated numerical instabilities."*

**\*Minor comments: - Figure 1, bottom row: add global-mean albedos somewhere on this figure?**

**- Page 7: the intercomparison fixes albedos, but what about sea ice dynamics?**

There is no dynamic ocean nor dynamic sea ice included since only ROCKE-3D is able to use such parameterizations.

**- Page 7: do the Hab1 and Hab2 cases with a zero-ocean-heat-transport slab ocean reach steady state on the nightside? In particular, does the sea ice thickness asymp- tote to a finite value?**

This is a level of details we reserve for the second part of the study when we will compare the outputs of each of the models.

**- Page 9 "should have reached radiative equilibrium" To what precision, in W/m2? Also, the global-mean top-of-atmosphere radiative equilibrium will be dominated by the warm dayside. The nightside could take much longer to reach equilibrium (smaller flux = longer equilibration timescale). Have the authors looked at the nightside surface bud- get, to see if it reaches equilibrium?**

This is an important question, however this question depends on the model itself. As described in Way et a.l, (2017), ROCKE-3D considers radiative equilibrium at a precision of 0.2 W/m$^2$. But other models may never reach such level of radiative equilibrium. Also, sometimes it requires other diagnostics such as surface temperature, sea ice extension, etc. to determine whether or

not the model reached convergence. Therefore, we prefer to remain agnostic concerning the threshold for radiative equilibrium and leave it up to the user to determine the convergence.

**- Page 11 "LMD-G is available upon request." From whom?**

We add: "LMD-G is available upon request from Martin Turbet (martin.turbet@lmd.jussieu.fr) and François Forget (francois.forget@lmd.jussieu.fr)"

**\*Technical comments: - Abstract, l.3: "... may soon be able to characterize, through transmission spectroscopy, the atmospheres of rocky exoplanets..." Why emphasize transits over other techniques that the manuscript mentions later on (e.g., emission spectra or phase curves)? The results of this work will be interesting more broadly.**

We modify this sentence into: *"through transmission, emission and reflection spectroscopy, the atmospheres of rocky exoplanets"*

**- Abstract, l.14 "The four test cases included two land planets composed of pure N2 and pure CO2, respectively..." ... pure N2 and pure CO2 \*atmospheres\*, ...**

Done

**- P2, l.6-7: "... and represent nearly 20% of astronomical objects in the stellar neigh- borhood of the Sun. " Interesting! Citation?**

it is actually 15 %: "Cantrell, J. R., Henry, T. J. & White, R. J. The solar neighborhood XXIX: the habitable real estate of our nearest stellar neighbours. *Astron. J.* **146,** 99 (2013). "

We have updated the 20 % to 15 % and added (Cantrell et al., 2013) as a reference.

**- P6, l.7: "because all the models do not include CO2 condensation" - because not all the models include X, or because all the models do not include X?**

"because not all the models include X" thank you.

**- P7, l.29: " to much the model" - typo, too.**

Done.

**- P7, l. 30: "disable the gravity waves" - the gravity wave parameterization in the strato-sphere? The dynamical cores should still be resolving some internal gravity waves.**

Yes, the gravity wave parameterization in the stratosphere (sub-grid). Thank you for pointing this out, we have modified the paragraph in:

*"We also ask the contributing scientists to disable the sub-grid gravity wave parameterizations in their model. Indeed, all the models do not have implemented a gravity wave parameterization and some have prescribed or predicted gravity wave formation, tuned for Earth topography and meteorology. Therefore, to avoid differences in atmospheric dynamics especially above the tropopause, we recommend to not include the sub-grid gravity wave parameterizations in this*

*intercomparison. Gravity waves whose wavelengths are greater than the model grid are explicitly resolved in the models and do not need to be modified."*

**- Page 8, Table 2: molecular air mass is referring to the dry background gases only?**

Yes only for dry gases, we add "(dry)" in the "molecular air mass" row of Table 2.

**- Page 8, Table 2: momentum roughness length and heat roughness length are missing units.**

Good catch. Both are in meter.

Done.

---

## Author Comment (AC2) · 14 Oct 2019

**This brief paper describes a protocol for inter comparing GCMs for TRAPPIST-1e in anticipation of future observations. The goal is to determine the differences in climate states when the models are run under similar configurations. These states can be related to spectra or thermal phase curves anticipated from future observations such as JWST. Four GCMs have signed on so far but only preliminary results are available; more detailed analysis will follow. This is a good idea and should be a useful effort.**

**The biggest uncertainty is the mass and composition of the atmosphere. The only constraint seems to be that HST observations do not favor an extended H2 atmosphere for TRAPPIST-1e. Thus, heavier atmospheres consisting mainly of N2 and CO2 are considered. The models are configured to sort out the effects of the dynamical core, physical packages, and moist processes. This is achieved by comparing four different runs for each model with different surface and atmospheric conditions. The approach seems reasonable and should give the authors a good start on what will surely be a challenging but stimulating research project.**

We would like to thank the reviewer for their positive words and their helpful comments, which have allowed us to make improvements to our manuscript. We have addressed each of the referee's comments below and noted the resulting changes we have made to the paper.

**The authors might consider a few things.**

**1. If the goal is to determine the differences in model climate states with similar run configurations, it is not clear how model numerics will be separated from model physics. Different dynamical cores running at different resolutions with different numerical schemes will produce different climates. How does one distinguish these differences from those due to real physical processes? I think BEN1/BEN2 should get at some of this but not all of it. Perhaps one way is to run with simple Newtonian cooling using a common relaxation field and time constant.**

We agree with referee #2 that a simulation with "Newtonian cooling" is an interesting idea. However, this increases the load of simulations to perform and we think that in order to have this intercomparison working in a reasonable time frame, the number of simulations should be low.

Also, the goal of this intercomparison is not to understand exactly why those models differ but to understand how these differences can have an impact on the observables from synthetic spectra. The four simulations we propose cover a large enough parameter space to answer this question. He have added the paragraph below before the conclusion:

*"Note that while additional simulations with a simple Newton cooling model, a 1-D column model, or with cloud radiative effects disabled would help to better understand the differences due to the dynamical cores and cloud physics, they will also dramatically increase the computational time, amount of data and effort. Yet, THAI aims to be easily reproducible and not time consuming in order to reach many GCM user groups. The five simulations propose in THAI should be enough to understand the main differences between the GCMs and their impact on the observable. THAI could also be used as a benchmark for a future GCM intercomparison that will specifically aim to understand each differences between the models."*

**2. Once that is clarified then an even more daunting task is to isolate changes due to different physics prescriptions. Is the intent to go to that level of detail or to describe what the differences are without analyzing the reasons? Some brief discussion about this would be helpful.**

What we are looking for with this intercomparison is the most important effects between the models, the ones that can have a first order impact on the climate. We therefore do not consider subtle effects due to the numerical schemes, resolution etc. As mentioned in our answer to referee #2 first question, the objective is to quantify the impact of the model differences on the observables, not to understand all the differences between the models which would require another experimental protocol.

**3. With suppressed CO2 condensation nightside surface temperatures are likely to be much warmer than when condensation is included. Without latent heat release atmospheric temperatures will cool and the surface must warm to maintain energy balance. Feedbacks related to moist processes may be affected and this may complicate the interpretation.**

We agree with referee #2 about the potential effect of disabling $CO_2$ condensation. However, this is something we have to set up in the future because not all the models include $CO_2$ condensation. For a similar reason we did not consider ocean heat transport (OHT). In the future, we hope that $CO_2$ condensation and OHT will be integrated in all the models.

**4. The radiative effects of clouds, both CO2 and H2O, can be very different between models. Runs with passive clouds might help isolate those effects. Of course, this adds to the analysis work (as does running with Newtonian cooling), but it is a point worth considering.**

Disabling the radiative effects of clouds has also been suggested by referee #1. We agree that this is a very interesting suggestion. However, as also mentioned in the answer of the first question of referee #2, this would increase the number of simulations required for the intercomparison. As an example, Hab1, Hab1* and Hab2 require about 65Gb of data each. Running the model with the radiative effects of clouds disable would require starting new simulations from the initial conditions. We believe that in order to be successful, the number of simulations in THAI should stay small (we already have 5) with the objective to stay focused on understanding the impact of the differences on the observable. A paragraph added before the conclusion discusses about this.

*"Note that while additional simulations with a simple Newton cooling model, a 1-D column model, or with cloud radiative effects disabled would help to better understand the differences due to the dynamical cores and cloud physics, they would also dramatically increase the computational time, amount of data and effort. THAI aims to be easily reproducible and not time consuming in order to reach many GCM user groups. The five simulations propose in THAI should be enough to*

*understand the main differences between the GCMs and their impact on the observables. THAI could also be used as a benchmark for future GCM intercomparisons that specifically aim to understand each differences between the models."*

**5. The authors hope to add more models into the mix which will increase the workload. Recognizing that this is not a proposal, it still begs the question of having adequate support and manpower to do the work. Is there?**

The first author, Thomas Fauchez, has a NASA SEEC proposal funded to work on this project with two THAI members as co-Is (Ravi Kopparapu, Mike Way). Therefore, we think that we have the adequate resources to successfully perform this intercomparison.

Also a THAI workshop is currently being planned around fall 2020 to discuss about THAI results and their perspectives (this is now mentioned in the revised version of the conclusion):

"The results of the comparison of these four models will be given in a second paper and a THAI workshop is planned for fall 2020."

---

## Author Response (AR2)

**Topical Editor Decision: Publish subject to minor revisions (review by editor)** (19 Nov 2019)
by Julia Hargreaves

We would like to thank the Topical Editor Julia Hargreaves for her careful reading of our manuscript and for her pertinent comments that significantly help to improve the manuscript. We have addressed all of them below.

Comments to the Author:
For a MIP model experiment description in GMD the following applies.
When we say "available" we mean either included as part of the paper or the supplement, or uploaded to a persistent public repository with a unique identifier.
Thank you for this comment.
The ROCKE-3D and ExoCAM are indeed available on a persistent public repository.
UM is also available on a persistent repository but for use under license.
Only LMD-G is not "available" in the sense that is used in GMD. Therefore we replaced: "LMD-G is available upon request from Martin Turbet…" into "LMD-G is obtainable upon request from Martin Turbet…"

1. The boundary conditions must be fully described and any required data made available.
I think this is fulfilled in Table 2
Yes

2. Baseline required model outputs should be indicated, with variables, length of averaging and length of output.
I think this is fulfilled in Table 3
Yes

3. There should be in place a mechanism for making the model outputs available.
It is less clear to me whether this has been fulfilled to the GMD standard.
a. When I click on the link in the Data availability section it does not resolve
Thank you for pointing this out, the link: https://thai.emac.gsfc.nasa.gov/dataset/thai was only available within NASA network but it is now available worldwild. Note that the repository will be needed for the second part of this project, when data from the GCMs will be uploaded.

b. Rather than appearing in the data availability section in passive voice, the information should be in the Outputs section and presented in the form of positive instructions to anyone wanting to participate.

We have now included this paragraph at the end of the output section:

"To facilitate comparison between each GCM, we ask the contributing scientists to provide their outputs in netCDF format.  The contributing scientist will be able to upload their data on a public permanent repository at https://thai.emac.gsfc.nasa.gov/dataset/thai after requesting an IP address authorization to Thomas Fauchez (thomas.j.fauchez@nasa.gov)."

Other things.

If you think there is any chance that you might change the protocol in any way while still remaining in the TRAPPIST-1 project, you should add a version number to the name of the protocol and include this in the title. Changes to the protocol cannot be written as an erratum to the paper, but must be submitted as a new paper, ie. these changes comprise are an update not an error.
This is a very good suggestion, we now added. "version 1.0" to the title.

Figure 1
Has it been adjusted as suggested by the reviewer or was the comment about there being no dynamic sea ice supposed to be an answer to the reviewer's suggestion?
We chose to not add "global-mean albedos" on this figure as suggested by reviewer #1 because i) it is very similar to the "reflected stellar" radiation of the third raw and ii) it will overload the figure.

Furthermore, this figure is meant as an illustration of preliminary results. We reserve more detailed and thorough analysis for the follow up paper.

Radiative equilibrium.
While it is OK if different model groups have different criteria, you should request any important criteria like this to be reported as part of the information about the model outputs. As a frequent user of MIP outputs I can tell you that it is a really good idea as part of the outputs to include information about each model, how it was set up, and a contact person who is prepared to answer questions about it. Some projects even get their members to write individual papers about how the runs were set up. Anyway, you should specify what you require in the new paragraph of the Outputs section of the paper where you are going to address the mechanism for making outputs available.

It was indeed a difficult question because the lowest value for the radiative imbalance could be very different depending on the model and its particular set up. Thus, we first decided to not include any threshold. However, following your recommendation we now consider the criteria described below and have added this paragraph in the output section:
"All the simulations should have reached radiative equilibrium at TOA at $\sim +/1$ $W.m^2$. If such limit can't be achieved by the model, the radiative equilibrium can be established if no discernible trend are observable in the last 10 year average global mean temperature."

Abstract

General Circulation Model
or
Global Climate Model
Pick one rather than pick and mix. :-)

" Global Climate Model" is more appropriated. We have replaced every "General Circulation Model" by "Global Climate Model" in the manuscript.

**Additional comment:**

Note that we have modified the Ben1 case by adding 400 ppm of $CO_2$ in addition of the 1 bar of $N_2$. That way, the protocol is symmetrical in term of 
[revised manuscript text omitted]